# Touching 4D Objects with 3D Tactile Feedback

Category: Research

## ABSTRACT

This paper introduces a novel interactive system for presenting 4D objects in 4D space through tactile sensation. A user is able to experience 4D space by actively touching 4D objects with the hands, where the hand holding the controller is physically stimulated at a three-dimensional hypersurface of the 4D object. When a hand is placed on the 3D projection of a 4D object, the force generated at the interface between the hand and the object is calculated by referring to the distance from the viewpoint to each point in the frontal surface of the object. The calculated force in the hand is converted to the vibration patterns to be displayed from the tactile glove. The system supplements the 4D information such as tilt or unevenness, which is difficult to be visually recognized.

**Keywords:** 4D space, 4D visualization, 4D interaction, tactile visualization

**Index Terms:** Human-centered computing—Visualization—Visualization application domains—Scientific visualization

## 1 INTRODUCTION

With the development of computer graphics technology, various methods for displaying 4D objects have been proposed. Furthermore, the VR technology allowed users to observe 4D objects with a higher degree of immersion. Some researchers produced new methods of visualization and interaction [1], [5], [9], being expected to improve the understanding and cognition of 4D space.

While most of these approaches use only the visual information by producing 4D space, humans also use auditory and tactile perception to recognize 3D representations. However, there are few studies focusing on these additional sensations for 4D spatial representations. In order to present richer four-dimensional information to users, auditory and tactile information is necessary as well as visual information. Therefore, we develop a novel system for displaying 4D objects using tactile sensations.

Needless to say, humans cannot touch objects defined in 4D space, since their bodies are situated in 3D space. The tactile sensations are the information mapped onto their skin, which is arranged in 2D surface. Therefore we can infer that the ones situated in 4D space touch 4D objects with their 3D skin, and the tactile sensation is regarded as the information mapped into 3D hypersurface, if it exists. Moreover, in general, humans' tactile perception appears to be mapped into their actual 3D space in combination with the body arrangement. Our idea is to introduce the experience as if humans touch the 4D objects by getting the 3D-mapped tactile-like information of 4D objects through their skin.

In this paper, we propose a novel 4D interaction system with tactile representation. We focus on the aspect of tactile sensation that can represent the unevenness of an object by displaying pressure sensation. We convert the tactile information into vibration patterns for ease of handling. For presenting the touch information, a tactile glove installing vibration actuators is developed.

The introduced system works as follows. First, a 4D object is projected to a 3D screen in the VR environment. Second, a hand wearing the tactile glove enters the screen. Third, the system calculates the necessary tactile information. Finally, the hand receives tactile sensations so that each part of the hand corresponds to each part of 3D hypersurface of the 4D hand. A user is able to perceive information of the projected 3D hyperplane such as the slope and unevenness as if he/she touches the 4D objects, in the sense that he/she

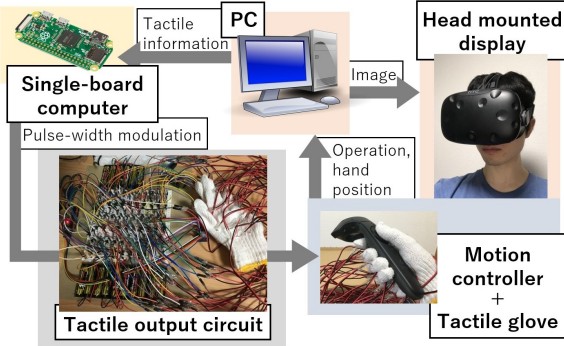

Figure 1: Overall view of the system.

receives 4D tactile-like information through the tactile sensation.

## 2 RELATED WORKS

Several approaches have been introduced for presenting 4D spatial information through tactile or haptic sensations. Hashimoto et al. [2] developed a method that visualized multidimensional data with haptic representation. They use a haptic device capable of 6-DoF input and force feedback to control a pointer floating in the virtual environment. When the pointer overlaps the displayed data, the value corresponding to the location is expressed in torque. The system also allows a user to explore four or more dimensional environment by operating the 3D slice with twisting input. Zhang et al. [7], [9] proposed a method using a haptic device for exploring and manipulating knotted spheres and cloth-like objects in 4D space. In the exploration of the objects, constraining the movement of the device to the projected objects improves the understanding of complex structure. In the manipulation of the objects, haptic feedback is presented by rendering the reaction of pulling force.

In the study of tactile presentation, Martinez et al. [3] proposed a haptic display by introducing a vibrotactile glove that enables users to perceive the shape of virtual 3D objects. Ohka et al. [6] developed a multimodal display, which is capable of stimulating muscles and tendons of the forearms and tactile receptors in fingers for presenting the tactile-haptic reality.

These approaches, however, are not intended for simulating 4D tactile stimulation.

## 3 SYSTEM CONFIGURATION

As shown in Figure 1, we develop a 4D visualization system consisting of a personal computer (HP, Intel Core i7-8700 3.20GHz, 8GB RAM, NVIDIA GeForce GTX 1060), a 6-DoF head-mounted display (HMD) with a motion controller (HTC VIVE), a single-board computer (Raspberry Pi Zero), and a tactile glove. A user wears the HMD, and holds the motion controller in the hand wearing the tactile glove. The user observes 3D-projected 4D space in the virtual environment through the HMD. The position of the hand is recognized by the motion controller. When the user touches a 4D object, the tactile glove presents the corresponding tactile sensation to the user's hand. The software is implemented with Unity 2018.3.3f1 and SteamVR Unity Plugin v2.2.0.

Figure 2 shows a tactile glove equipped with a total of 30 vibration motors. Five motors are arranged on the palm side of the hand, five

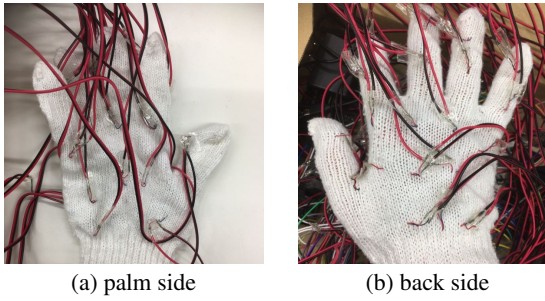

Figure 2: Tactile glove.

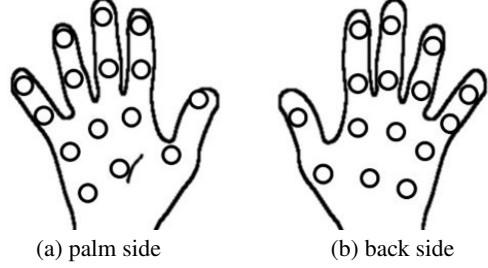

Figure 3: Arrangement of vibration motors in tactile glove.

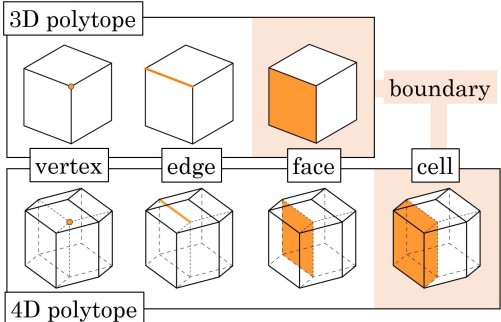

Figure 4: Structure of the 4D polytope compared to the 3D polytope.

on the back side, and two on the front and the back of each finger as shown in Figure 3. The motors are driven by electric current, and the strength of the vibration stimuli is controlled by the single-board computer with pulse-width modulation.

## 4 4D VISUALIZATION SYSTEM

In this section, we describe the 4D visualization system which projects 4D objects into 3D virtual environment. The system is based on an algorithm introduced in McIntosh's 4D Blocks [4].

### 4.1 Definition of 4D objects

In the system, we define 4D objects as 4D convex polytopes. Non-convex polytopes are constructed by combining convex polytopes. As shown in Figure 4, boundaries are composed of 3D objects called cells, and their intersections are divided into 3 different features; vertices, edges, and faces, depending on the number of dimensions.

### 4.2 Projection

Figure 5 shows the 4D projection model. 4D objects arranged in 4D space are projected to a 3D screen. Positions of 4D objects are described by 4D vectors in the 4D coordinate system, and their orientations are described by $4 \times 4$ orthogonal matrices. The center

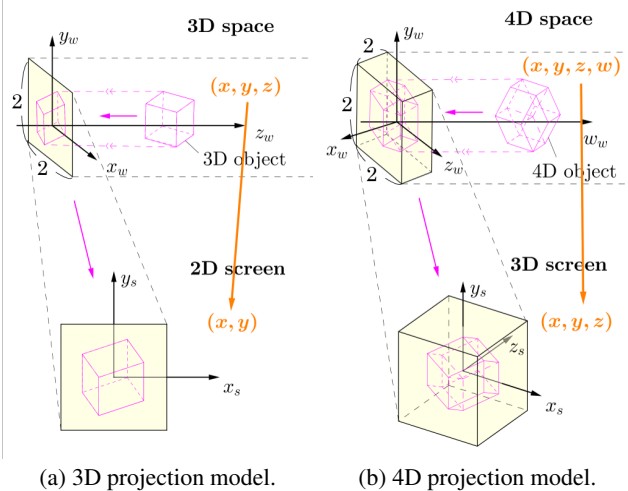

(a) 3D projection model.      (b) 4D projection model.

Figure 5: 3D and 4D projection model.

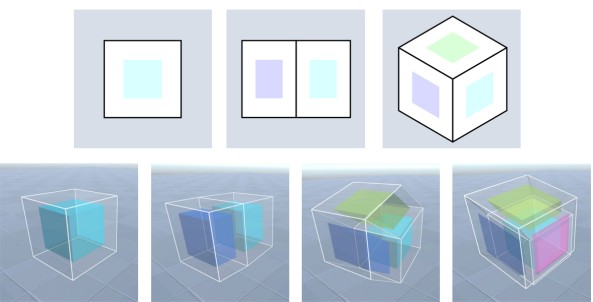

Figure 6: Examples of drawings with similar drawings in 3D.

of the 3D screen is located at the origin of the 4D space. The 3D screen has a dimension of $2 \times 2 \times 2$, spreading in $x_w y_w z_w$ plane. Data defined in the 4D world-coordinate system $x_w y_w z_w w_w$ are converted to data in the 3D screen-coordinate system $x_s y_s z_s$ by removing $w$-coordinated component. 4D objects are orthogonally projected in the 3D screen by this transformation. From the viewpoint of the suitability of tactile presentation, we selected the orthographic projection method. The system also supports perspective projection, which can be occasionally selected by a user.

### 4.3 Drawing

In the system, face polygons and edge lines are used for drawing. Polygons and lines do not have normals in 4D space, and are drawn as contours of cells. In order to make it easier to distinguish cells displayed on the 3D screen, the system draws the center of cells with semi-transparent color-coded polygons, in addition to edges and faces. Figure 6 shows the examples of drawings.

For accurate drawing, back-cell culling and hidden-hypersurface removal are applied. Occluded cells are determined by taking the dot product of a view direction and cell normals, and faces that belong to visible cells are selected so that they will be used for drawing.

As the system doesn't adopt voxel rendering [1], we cannot use depth buffering for hidden-hypersurface removal. Instead, we use *clip units*, which represents an area where an object obscures other objects behind it. Figure 7 shows an example scene where clip units work. As shown in Figure 8, clip units are identified by contact subsurfaces of front-facing surfaces and back-facing surfaces with respect to the viewer. Each clip unit is a halfspace whose boundary includes the subsurface, and the boundary is orthogonal to the screen. Intersections of clip units are used for clipping. Drawing polygons

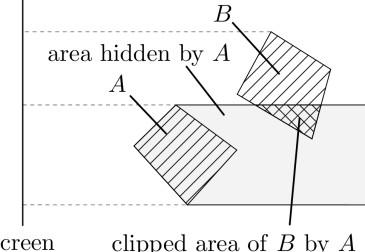

Figure 7: An example scene where clip units work. We don't draw the intersection of rectangle B and the area hidden by A.

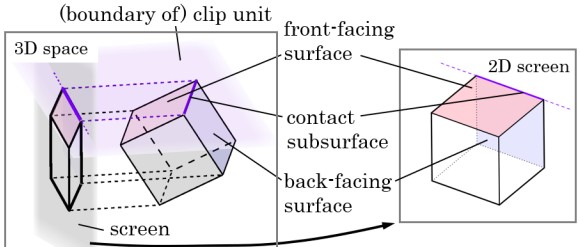

Figure 8: Calculating a clip unit in a 3D scene. Note that clip units are dimension-independent concepts. In $n$D, "surfaces" mean $(n-1)$D components, and "subsurfaces" mean $(n-2)$D components.

and lines are clipped by clip units of objects closer to the viewer.

### 4.4 Control

4D orientations have six degrees of freedom [8]. A user is able to rotate a 4D object by moving the motion controller while pressing the trigger. To naturally correspond the controller's 6-DoF operation in 3D space with the object's 6-DoF rotation in 4D space, the 3-DoF translational operation (Figure 9(a)) is related with the rotation involving the $w$-axis, and the 3-DoF rotational operation (Figure 9(b)) is associated with the rotation not involving the $w$-axis.

## 5 TACTILE REPRESENTATION

In this section, we introduce a tactile representation method. For explaining the concept, we firstly describe how to express the touching sensation of a 2D-projected 3D object, and then expand it to the 3D projection of a 4D object.

### 5.1 Analogy of touching 3D objects in 2D space

When humans touch a screen where a 3D object is displayed, they feel the object as if it is situated. For example, suppose that we move the right palm straight towards the front wall. Figure 10 depicts three different situations. When the wall faces straight in front, a uniform pressure will be applied on the palm (Figure 10(a)). Alternatively, when the wall faces a little to the right, the thumb will first be stimulated. When the hand is pressed against the wall as it is, the wrist will bend and the entire palm will touch the wall, but the thumb side will receive a stronger force than the little finger side (Figure 10(b)). When the hand touches a corner of the wall, stronger sensation is perceived at the center of the hand (Figure 10(c)). These differences can be distinguished by the pressure applied to a palm.

The calculation of tactile stimulus generation proceeds as follows. We consider 3D space where a 3D object is situated, and the space is displayed in a 2D screen. When a user wearing a tactile glove touches the screen, hand-touched area is projected on the surface of the object. Then the distance between the projected hand and the screen is calculated. Figure 11 shows the three different situations of a wall, related with the cases presented in Figure 10. If the distance to the left is shorter than that to the right, it means the wall faces

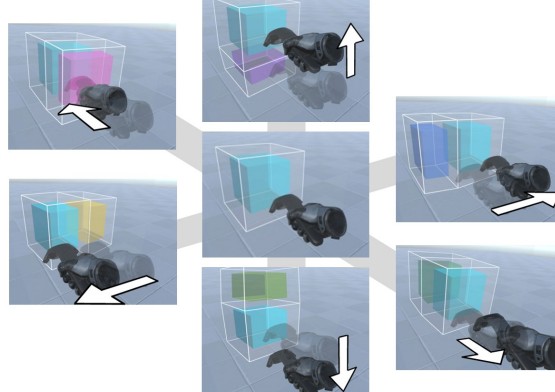

(a) Translation operation.

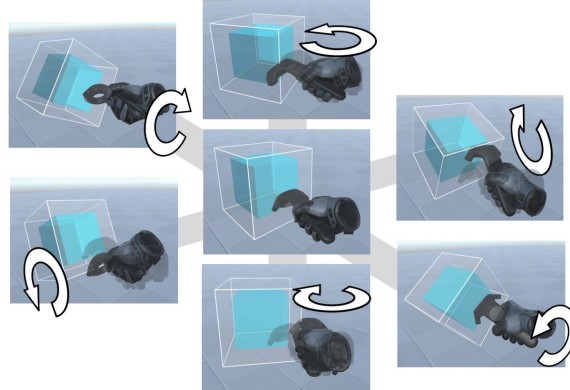

(b) Rotation operation.

Figure 9: Rotating 4D objects by the motion controller operation.

to the right. In this way, the strength of pressure is calculated by referring to the relative relationship of the distances. Figure 12 shows the results of the calculation.

Let $\Omega$ be a set of all actuators installed in the tactile glove. First of all, the actuators are projected orthogonally through the screen, and a subset $L \subset \Omega$ of actuators being projected on the object is detected. For each projected points of actuators $i \in L$, the distance $l_i$ from the screen is calculated. Then the normalized relative strength $s_i$ is calculated:

$$s_i = \max\{\alpha(l_{min} - l_i) + 1, 0\}, \tag{1}$$

where $l_{min} = \min_{k \in L} l_k$ and $\alpha$ is a gradient constant. The formula is derived as shown in Figure 13.

In general, corners give stronger pressure than a flat wall, and apexes give stronger pressure than corners. In the same way, an uneven wall give stronger pressure than a flat wall. By considering this fact, the strength of the stimulus $h_i$ is adaptively adjusted according to the degree of sharpness and inclination:

$$h_i = \left(1 - \beta \frac{\Sigma_{k \in L} s_k}{N}\right) s_i, \tag{2}$$

where $\beta$ is a reducing constant ($0 \le \beta \le 1$) and $N = \#\{i \in L \mid s_i > 0\}$ is the number of active actuators. Figure 14 shows examples of the application of the formula.

Finally, the strength $h_i$ is converted to voltage value $V_i$:

$$V_i = \gamma h_i, \tag{3}$$

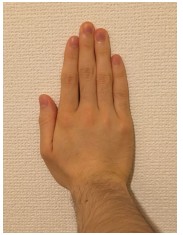 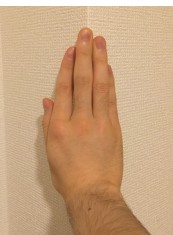

(a) Wall facing straight in front.

(b) Wall facing to the right.

(c) Corner of the wall.

Figure 10: Various situations when touching a wall.

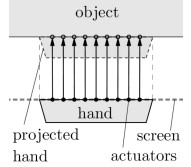 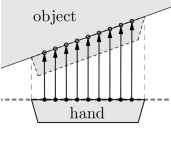 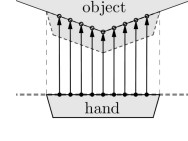

(a) Wall facing straight in front.

(b) Wall facing to the right.

(c) Corner of the wall.

Figure 11: Calculation of tactile stimulus generation.

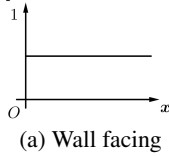 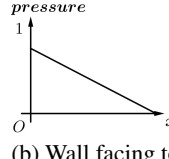 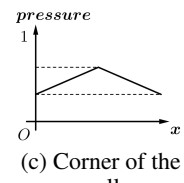

(a) Wall facing straight in front.

(b) Wall facing to the right.

(c) Corner of the wall.

Figure 12: Calculation results.

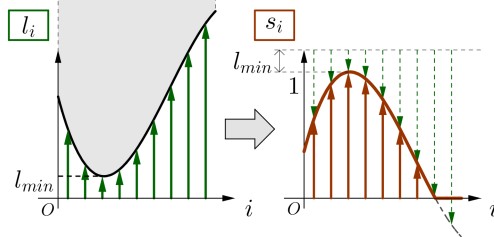

Figure 13: Visualization of equation 1. ($\alpha = 1$)

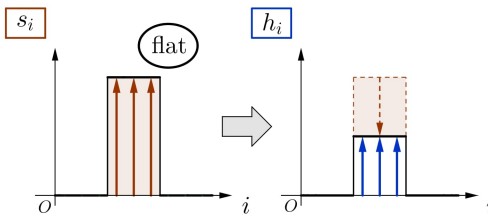

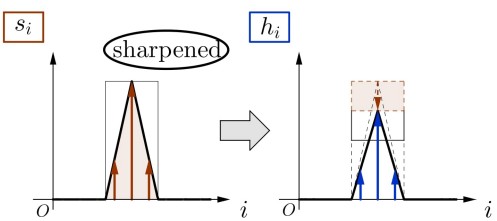

Figure 14: Visualization of equation 2. ($\beta = 0.5$)

where $\gamma$ is a constant.

## 5.2 Applying to 4D system

The idea and the calculation method presented in the previous section can be extended to the 3D projection of 4D objects. In the 3D system, a user receives 2D-mapped stimulus patterns through his/her palm of the hand (Figure 15(a)). In the 4D system, the user receives 3D-mapped data through all the skin of the hand holding the controller (Figure 15(b)). Touching the 2D screen in the 3D system corresponds to putting a hand in the 3D screen in the 4D system. When the hand enters the screen, the stimulus is calculated in the same way as the 2D cases.

Figure 16 shows the model of tactile calculation. The VR environment of the system consists of the 3D screen and a rendered user's hand. 4D object is kept at a distance with the screen so that the hand prevents from colliding with the screen. When the hand enters the defined 4D space, the relative position of each actuator from the screen $d_i = (x_{d_i}, y_{d_i}, z_{d_i})$ is detected in the 3D screen coordinate system $x_s y_s z_s$, and the position is converted to the 4D world coordinate system $x_w y_w z_w w_w$ as

$$c_i = (x_{c_i}, y_{c_i}, z_{c_i}, w_{c_i}) = (x_{d_i}, y_{d_i}, z_{d_i}, 0). \quad (4)$$

Then the line $\mathbf{L}_i$ extending from $c_i$ towards positive direction of $w_w$ axis is defined:

$$\mathbf{L}_i = c_i + (0, 0, 0, t) \ (t \geq 0). \quad (5)$$

By clipping $\mathbf{L}_i$ with the 4D object, the intersection point

$$q_i = (x_{q_i}, y_{q_i}, z_{q_i}, w_{q_i}) = (x_{c_i}, y_{c_i}, z_{c_i}, t') \quad (6)$$

of $\mathbf{L}_i$ and the object is detected. If $\mathbf{L}_i$ isn't clipped by the object, the corresponding location of the user's hand is not projected on the object.

Here, $l_i$ is calculated as the distance between $c_i$ and $q_i$:

$$l_i = \sqrt{(x_{c_i} - x_{q_i})^2 + (y_{c_i} - y_{q_i})^2 + (z_{c_i} - z_{q_i})^2 + (w_{c_i} - w_{q_i})^2}. \quad (7)$$

$l_i$ is calculated for all locations, and converted into the strength of the stimulus by referring to the equations (1), (2) and (3).

## 6 RESULTS OF TACTILE DISPLAY

The system is able to display multiple objects from any viewpoint, however in this section, we deal with a single object situated in the center of the screen for a simple example.

We implemented a function to visually display the strength of the stimulus calculated at each location for visually validating the operation. For precise rendering of tactile stimuli, the calculation is conducted at 242 locations arranged in grids situated on the user's hand. As shown in Figure 17 (a), (b), and (c), the locations and the amplitude of each tactile stimulus are imposed on the left side of the 3D screen. The displayed stimuli in a cube show the area to be mapped on a virtual hand, so that the user is able to intuitively recognize the presented tactile sensation given to the hand. Based on the 242 calculated values in the cube, the stimuli at 30 points corresponded to the motor locations, which is colored in cyan, are simultaneously given to the motors for presenting tactile sensation.

As shown in Figure 17, when a user touches a hypercube with its cell facing the front, a uniform stimulus is generated. The magnitude of this stimulus is the same, even when only part of the hand is touched. Note that in all the following figures, the hand is in the same orientation.

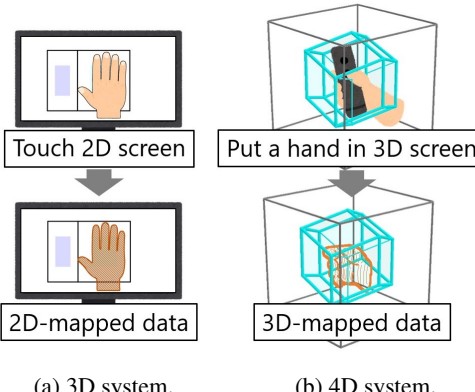

(a) 3D system.      (b) 4D system.

Figure 15: Tactile system model.

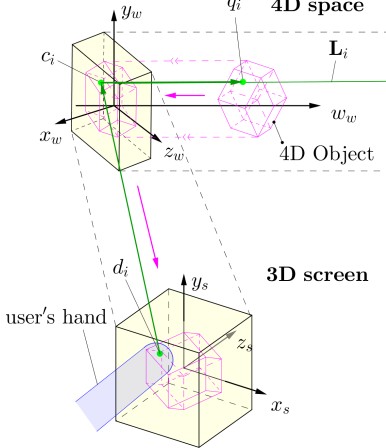

Figure 16: Tactile calculation model.

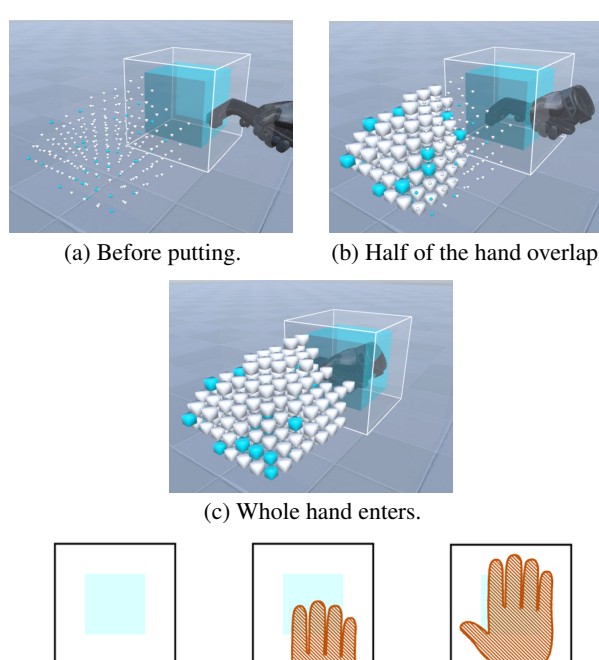

(a) Before putting.      (b) Half of the hand overlaps.

(c) Whole hand enters.

(d) 3D analogues for the above three cases.

Figure 17: Putting a hand inside the cube representing the surface of a hypercube.

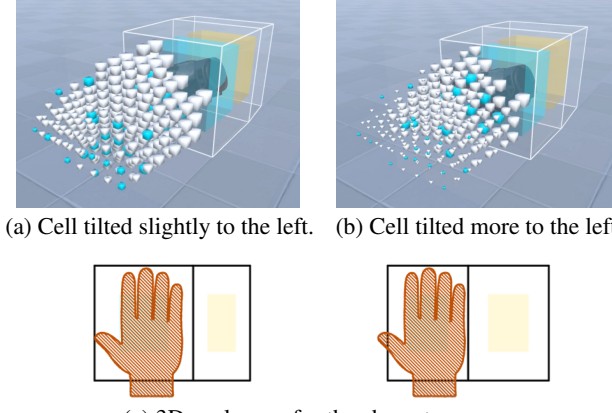

(a) Cell tilted slightly to the left.    (b) Cell tilted more to the left.

(c) 3D analogues for the above two cases.

Figure 18: Touching the hypercube tilted slightly to the left.

If the cell is tilted slightly to the left, the stimulus becomes greater on the right and smaller on the left (Figure 18(a)). The gradient increases as the slope increases, and eventually the leftmost stimulus disappears (Figure 18(b)).

By manipulating the hypercube, the user is able to touch and recognize the shape intuitively. When the face is facing the user, the wall-shaped tactile sensation is presented as shown in Figure 19 (a). If the edge is situated in the front, the user feels line-shaped sensation as presented in Figure (b). The vertex presents an isolated strong stimuli, and the sensation gets gradually weaker in the peripheral area as shown in Figure (c).

The user is also able to recognize differences between objects which look the same. Three different objects depicted in Figure 20 (b), (c), (d) are observed as the same from certain directions (Figure 20(a)). The three objects are recognized differently by the touch sensation. Figure 21(a) presents a 4D cone, where the centered area gives strong stimuli, and the stimuli gradually decreases in the perifieral area. When touching flat surface, the stimuli appear uniformly as shown in Figure (b). For the hollow shape, the stimuli in the central area is weaker than the surrounding area as presented in Figure (c).

The tactile system was experienced and evaluated by one subject. Tactile stimuli rendered by vibration patterns were correctly presented as designed, and the subject could distinguish the above differences as well. He had a little difficulty to recognize the exact boundary of faces by different stimuli, however it could be compen-

sated by moving the hand appropriately.

## 7   CONCLUSION AND FUTURE WORK

In this paper, we proposed the system to display the 4D shape by rendering the 4D tactile sensation. 4D cognition can be supplemented by combining the visual information with the tactile information. By increasing subjects, the system should be verified objectively and quantitatively in future work.

Possible improvements are considered. Since this system uses simple vibration motors for tactile rendering, the resolution of strength and position is not enough for subjects to recognize more detailed shapes. The choice of high-quality actuators will solve this problem. Moreover, finely-controlled stimuli may be able to express much rich information such as friction and directional force.

The 4D space can be enriched by introducing four-dimensional physics. In the actual 3D world, an object moves by collision ac-

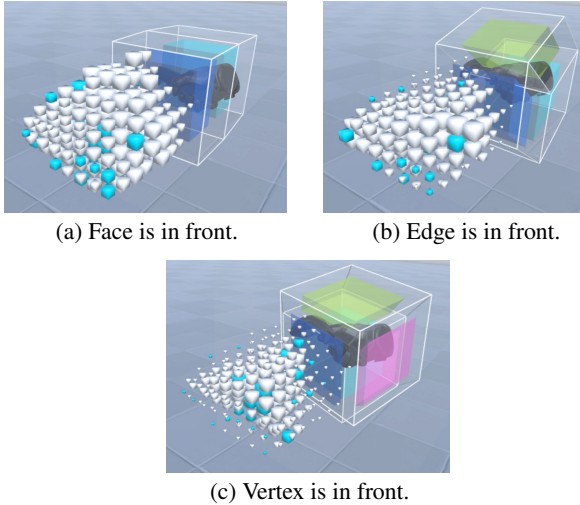

(a) Face is in front.          (b) Edge is in front.

(c) Vertex is in front.

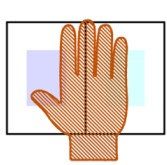
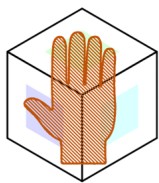

(d) 3D analogues.

Figure 19: Touching face, edge and vertex of the hypercube.

cording to the laws of physics. By combining with 4D physics, the tactile experience will be much realistic.

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

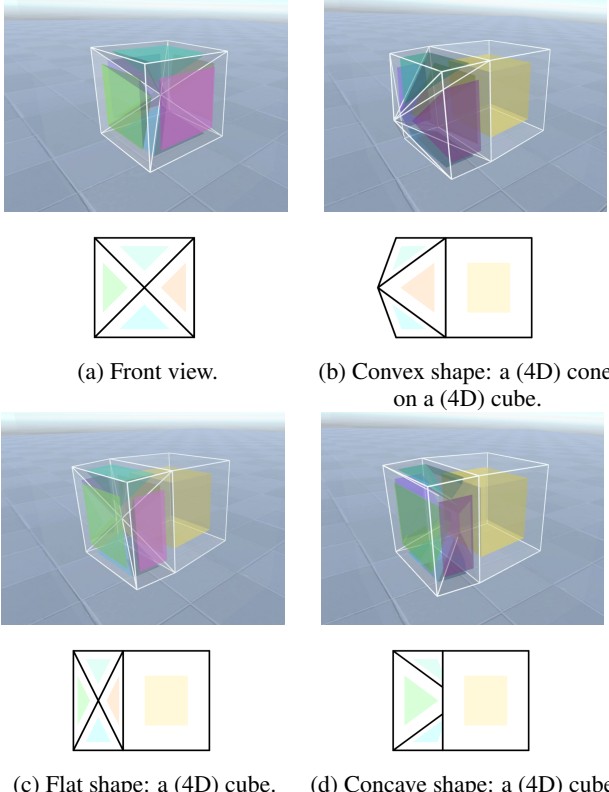

(a) Front view.          (b) Convex shape: a (4D) cone on a (4D) cube.

(c) Flat shape: a (4D) cube.          (d) Concave shape: a (4D) cube with (4D) cone hollowed out.

Figure 20: 4D objects and their 3D analogues.

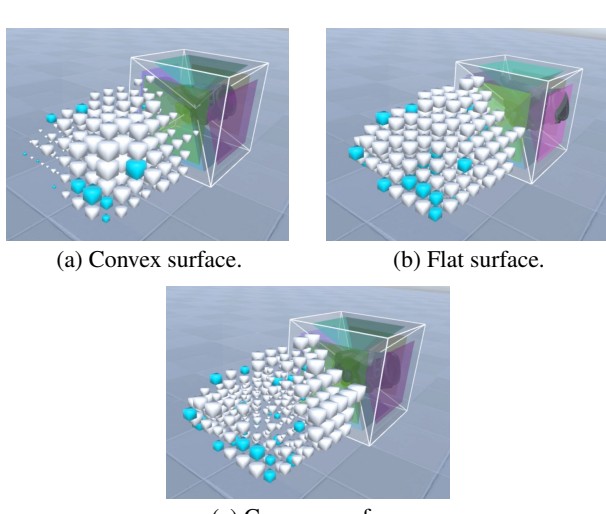

(a) Convex surface.          (b) Flat surface.

(c) Concave surface.

Figure 21: Touching 4D objects that looks the same when seen from the front.

