# OpenReview forum: "Touching 4D Objects with 3D Tactile Feedback"
_graphicsinterface.org/Graphics_Interface/2021/Conference/Second_Cycle — Reject_

### Official Review · Reviewer_y8mc · 2021-05-03
**3D simulation of 4D objects - using a haptic glove**

**Rating:** 4
**Confidence:** 3

**Review:**

This paper presented a method for enabling users to explore 4D objects using haptic feedback. The real key of the work is the interesting mapping approach given used to map the 4D data into the 3D spatial world (using the haptic feedback as the 4th dimension). The authors used the idea of pushing a flat hand onto a non-flat object (a 3D->2D projection), to inspire an analogous 4D->3D mapping. I find this very interesting.

Overall, however, I found the paper very hard to read. I am familiar with graphics and 4D objects, but really struggled.
- try to not mix implementation and theory as much, as it makes it more difficult to follow.
- clarify early on that you are working in actual 4D space (e.g., tesseracts, etc.) and not 3D + other dimensions

I really struggled to get my head around your mapping, and, I still think it's underspecified. How do you define the "frontal" surface in 4D space? What w value do you give to the hands' positions to help map it into the 4D space (necessary for your distance calculations)? How do you justify a whole-hand reaction haptic pattern when the hand itself will be in 3D (assuming it's not flat) vs a per-node reaction? And so forth.

Finally, I thought the work would have been better motivated by explaining the limitations with existing approaches. E.g., focusing on 4D->3D mapping instead of development of haptic devices.

So in all, I think there is a really cool gem in this work with the 4D->3D mapping approach, but I don't feel that it's presented in a way that will help people understand it or develop from it.

---

### Official Review · Reviewer_ume5 · 2021-05-03
**The paper seems decent, but questionable application area.**

**Rating:** 6
**Confidence:** 2

**Review:**

The paper introduces a novel trial of representing a 4D object into 3D space with an additional dimension of the haptic display. The analogy of 3D to 2D helped a general understanding of the 4D to 3D projection mapping, and the rationale behind the haptic cue seems to be well-founded theoretically. The figure 17 through 21 are especially good visualization of how the system works.

However, I have a question about the practical application of the implemented system, which is not clear in the paper. I can be convinced that a 4D object could be projected into a 3D world with the proposed system; however, what's the benefit of this? Does it improve the understanding of 4D space to a learner or a user? The evaluation part is extremely brief with only one subject and provides almost no information. A follow-up discussion about the system's empirical value, especially for the novice to this topic, is needed to be accepted.

Disclaimer: I have a limited understanding of 4D object representation and manipulation.

---

### Official Review · Reviewer_fnra · 2021-05-04
**Interesting concept, but necessary improvements in the presentation**

**Rating:** 5
**Confidence:** 3

**Review:**

The paper presents a novel approach for displaying 4D objects with tactile sensations. The tactile information is brought through vibration on gloves with front and back side motors. The core idea in the paper seems interesting and practical. However, the presentation needs to be largely improved. Both the writing, the implementation, and the example figures need to be more clear. Therefore, the recommendation is leaning towards rejection.

In particular, it is not clear in the beginning what the 4th dimension is. Try to clarify what the main goal of the paper is early in the abstract and the introduction, so non-experts in the field of 4D visualizations can follow the content. Furthermore, a stronger motivation of the benefits and contributions of the method is missing in the introduction.

It might help the understanding of the method if section 5 (Tactile Representation) was presented earlier in the paper.

Figure 5 is not well explained. It would be good to have some more intuitive explanation in writing and more details on what is presented in the figure.

Figures 6 and 7 are also unclear. More explanation, especially important for people who are not experts in 4D visualization, is necessary.

Can you include interaction between 2 objects in 4D space? It would be interesting to see an example like that.

Typo on page 2, section 4, “introdued”.

---

### Meta-Review · Area_Chair_SSUc · 2021-05-05

**Recommendation:** Reject
**Confidence:** 3

**Metareview:**

The paper introduces an interesting new idea for visualizing 4D objects using a head-mounted display and tactile glove. However, the quality of the paper presentation needs to be substantially improved. Hence, the recommendation is to reject the paper in the current form. Key concerns raised by the reviewers include the clarity of the exposition, figures, and examples, and the questionable application areas for the proposed method.

---

### Decision · Program_Chairs · 2021-05-08

Reject